# Magnesium Administration in Chronic Kidney Disease

**DOI:** 10.3390/nu15030547

**Published:** 2023-01-20

**Authors:** Emma A. Vermeulen, Marc G. Vervloet

**Affiliations:** Department of Nephrology, Amsterdam UMC, De Boelelaan 1117, 1081 HV Amsterdam, The Netherlands

**Keywords:** magnesium administration, magnesium supplementation, magnesium, chronic kidney disease, cardiovascular disease

## Abstract

Awareness of the clinical relevance of magnesium in medicine has increased over the last years, especially for people with chronic kidney disease (CKD), due to magnesium’s role in vascular calcification and mineral metabolism. The inverse association between serum magnesium and clinically relevant, adverse outcomes is well-established in people with CKD. Subsequent intervention studies have focused on the effect of magnesium administration, mainly in relation to cardiovascular diseases, mineral bone metabolism, and other metabolic parameters. The most commonly used routes of magnesium administration are orally and by increasing dialysate magnesium. Several oral magnesium formulations are available and the daily dosage of elemental magnesium varies highly between studies, causing considerable heterogeneity. Although data are still limited, several clinical studies demonstrated that magnesium administration could improve parameters of vascular function and calcification and mineral metabolism in people with CKD. Current clinical research has shown that magnesium administration in people with CKD is safe, without concerns for severe hypermagnesemia or negative interference with bone metabolism. It should be noted that there are several ongoing magnesium intervention studies that will contribute to the increasing knowledge on the potential of magnesium administration in people with CKD.

## 1. Introduction

Magnesium (Mg) is the fourth most abundant cation in the human body, following sodium, potassium, and calcium. Compared to these minerals, Mg has frequently been neglected in medicine, although attention to its clinical relevance increased last years. The majority of Mg resides intracellularly and in bone as a component of minerals. Only a minor fraction of total body Mg is present in extracellular fluids, with less than 1% of total Mg found in serum. Approximately 30% of serum Mg is protein-bound, and the remainder circulates as free ionized (up to 70%) or as mineral complex (less than 10%) [1]. Mg homeostasis involves dietary intake, subsequent (partial) absorption, and renal excretion (Figure 1) [2]. A variable fraction of unabsorbed dietary Mg is also excreted through the gastrointestinal tract.

Intestinal Mg absorption is mainly, but not entirely, passive, through paracellular route, with a variable bioavailability of 20–80% [1,2,3,4]. This partial Mg absorption is not directly proportional to Mg intake, but also dependent on the bodily Mg status. Intestinal Mg absorption is also regulated by active transcellular transport across Mg channels, transient receptor potential channel melastatin member (TRPM) 6 and TRPM7, which are present primarily in the distal ileum and the colon [1,3]. Some Mg uptake occurs at the jejunum, where levels of the active vitamin D, 1,25-dihydroxy vitamin D3, are of relevance for Mg absorption in people with CKD [5]. This study demonstrates that jejunal Mg absorption increases after supplementation of 1,25-dihydroxy vitamin D3 in people with end-stage kidney disease-associated vitamin D deficiency. Serum Mg concentration is primary regulated through renal Mg excretion, with high tubular reabsorption (approximately 95%) and only 3–5% of ultrafiltered Mg being excreted by the kidneys [6,7,8]. With decreasing kidney function, serum Mg slightly increases because of reduced glomerular filtration, to some extent compensated by impaired tubular reabsorption [8,9,10]. In addition to this exchange with the milieu exterior, Mg exchange exists between compartments such as bone (which contains up to 60% of total bodily Mg), muscle (up to 30%), and other soft tissues (up to 20%) [2]. This Mg homeostasis is dependent on various endogenous factors, including parathyroid hormone (PTH), insulin, aldosterone, and estrogen levels [1,2]. In addition, Mg homeostasis is also influenced by external factors, such as commonly used drugs including proton pump inhibitors (PPI) and diuretics, through inhibition of active intestinal absorption and interference with urinary Mg handling, respectively [2,11,12,13,14].

Although measurement of total serum Mg is the most widely used and a convenient way to estimate bodily Mg status, this estimate has limitations because it is based on the assumption that a steady state exists between Mg compartments and the proportion that is protein-bound. However, this is not the case and it varies over time due to its hormone sensitivity and a possible circadian rhythm [15]. An early observational study already concluded that serum Mg does not reliably reflect intracellular or total body Mg content [6]. Considering that serum Mg comprises only a small fraction of total body Mg, various other methods of Mg measurement have been explored, among other intracellular Mg, i.e., erythrocyte or lymphocyte Mg, muscular Mg, and metabolic assessment through 24-hour (24-h) urinary excretion [1,16,17,18]. Somewhat contradicting the assumption that serum Mg does not adequately reflect total Mg content are findings from a Dutch population-based cohort study, where different methods of assessing intra-erythrocyte Mg correlated strongly with serum Mg levels as well as 24-h urinary excretion [17]. In turn, the lack of a consistent relation between Mg content of compartments and its serum concentration is exemplified by the observation that supplementing Mg does not necessarily change serum Mg concentrations. This suggests that Mg is preferentially directed into cells and tissue, thereby maintaining stable serum Mg levels, if it is not excreted directly. Since Mg measurement in serum and plasma is considered interchangeable, this review will only use the term ‘serum Mg’ for reasons of consistency and readability. The reference range of serum Mg is based on prevalent Mg concentrations in the general population and generally set at 0.70–1.05 mmol/L [4]. However, based on observational studies in populations with chronic kidney disease (CKD) including those treated by dialysis, it is suggested that for people with CKD, higher Mg levels should be aimed for [19,20]. Following these findings, several intervention studies have been conducted in recent years, using various ways to increase Mg in people with CKD.

In this review, we will describe studies assessing the effects of Mg administration on intermediate and clinically relevant outcomes in people with CKD stages 1 to 5. These outcomes include vascular calcification, cardiovascular disease (CVD), mineral bone disease (MBD)-related outcomes, inflammation, and mortality. Moreover, we will provide an overview of different routes of Mg administration and its effect on Mg homeostasis. We will also address the concerns related to Mg administration in CKD and present our position on the potential of Mg administration for individuals with CKD.

## 2. Modes to Administer Magnesium

The most common routes of Mg administration in people with CKD are oral supplements, changing dietary patterns to Mg-enriched products, and increasing dialysate Mg concentration for those treated with hemodialysis or peritoneal dialysis. Several oral Mg formulations are available, including Mg citrate, Mg oxide, Mg hydroxide, Mg sulfate, and Mg carbonate. Mg citrate has the best bioavailability compared to other formulations [21], and Mg carbonate is most frequently prescribed in hyperphosphatemia due to its phosphate-binding capacity [21,22,23,24]. The dose of oral Mg supplementation varies highly between studies, ranging from 49 and 729 mg (2–30 mmol) elemental Mg per day (Table 1) [22,23,24,25,26,27,28,29,30,31,32].

To optimize absorption and to overcome dose-limiting laxative effects of Mg supplements, it is recommended to distribute the total daily dose of Mg with intake twice or three times a day. Considering that the recommended dietary allowance (RDA) by the European Food Safety Authority and United States Food and Nutrition Board is 300–320 mg and 350–420 mg/day in adult women and men, respectively [33,34], Mg supplements commonly double dietary Mg intake. Dietary Mg intake results mainly from green vegetables, nuts, seeds, and unprocessed/whole grains and to a lesser extent from fruits, fish, and meat. It is important to realize that the mineral content of vegetables has drastically decreased over the years due to modern farming techniques and that processed food is generally low in Mg content [16]. The natural mineral content in drinking water differs highly (based on type of water purification systems), with a mean European tap water Mg content of 9.6mg/L [35,36]. To our knowledge, no studies have investigated Mg administration through increased Mg concentration in drinking water, although a few studies investigated the association with tap water Mg concentration in relation to CVD and mortality, as will be discussed later [36,37]. Dietary intervention studies in humans are difficult to perform and notoriously difficult to interpret, since dietary modifications will inevitably also lead to changes in intake of multiple nutrients, making it impossible to ascribe potential beneficial observed effects to Mg only. Therefore, human dietary Mg studies are generally observational. However, dietary intervention studies focusing on isolated incremental dietary Mg intake have been performed in rodents with kidney failure, such as 5/6 nephrectomised rats and in Klotho knock-out mice [38,39,40,41,42]. These experimental studies all demonstrated that a high-Mg diet compared to a low-Mg diet prevented or retarded the development of vascular calcification.

During routine dialysis, a dialysate Mg concentration of 0.50 mmol/L is commonly used, generally leading to an intradialytic decline in serum Mg concentration [43]. An increase of the dialysate Mg concentration equal to or above 0.75 mmol/L or 1.0 mmol/L is an effective way to supplement Mg, leading to a rapid increase of post-dialysis serum Mg [44,45,46]. Most studies focused on Mg administration through haemodialysis. Other routes of Mg administration, such as long-term intramuscular and subcutaneous Mg administration, have been described in case reports and demonstrated to effectively restore Mg deficiency, but cause local pain and irritation [47,48,49]. When using a dilution of Mg in combination with the addition of lidocaine 2%, these symptoms can be tolerable [47]. The possibility of transdermal Mg administration through Mg-containing salt baths, oils, or creams have been suggested; however, scientific literature is very limited and studies are small. However, these unconventional methods are not considered suitable for routine clinical practice if other routes of Mg administration are feasible.

## 3. Magnesium Administration to Improve Health

Several lines of evidence suggest that restoration of Mg deficiency, or even supplementing Mg in the absence of overt hypomagnesemia, may promote health or retard the progression of pathological conditions or diseases. For this reason, a number of studies have been performed, aiming to prove the benefit of Mg administration on various outcomes, which will be addressed in the subsequent sections. An overview of results that support a rationale for Mg administration is presented at the end of these sections (Table 2).

### 3.1. Impact of Magnesium Administration on Cardiac Function

On a molecular level, Mg is extensively involved in a wide variety of enzymatic processes with a key role as a co-factor for many adenosine triphosphate (ATP)-dependent reactions. In addition, Mg affects the resting transmembrane potential, regulates the function of several ion channels that are involved in cell repolarization, influences cardiac remodeling, and is essential for cardiomyocytes relaxation [1,19]. Therefore, Mg is a crucial element in the complex mechanism of cardiac function. Many studies investigated the association between serum Mg and the prevalence of cardiac disease, including arrhythmias, sudden cardiac death, or heart failure (HF) in people with and without CKD [51,52,53,55,56,57,58,59,84]. Although most of these studies concluded that lower serum Mg levels are associated with an increased risk of arrhythmias, sudden cardiac death, and HF, subsequent intervention studies assessing the effects of Mg administration on these cardiac outcomes are sparse. To our knowledge, there are no intervention studies on the effect of Mg administration on arrhythmia, sudden cardiac death, or heart failure in CKD. A small, cross-over intervention study in people on hemodialysis reported that an Mg dialysate concentration of 0.25 mmol/L, compared to 0.50 mmol/L resulted in lower pre- and post-dialyses serum Mg and a higher prevalence of ventricular arrhythmias [54]. In the setting of cardiac surgery on congenital heart disease, Mg administration has been investigated to prevent post-operative arrhythmias [84,85]. A propensity score-matched, prospective cohort study demonstrated that Mg administration (25 mg/kg–50 mg/kg) through the cardiopulmonary bypass circuit was associated with approximately 50% reduction for post-operative occurrence of various arrhythmias [84]. A dietary Mg intervention study in Sprague Dawley rats with normal kidney function, an Mg-deficient diet resulted in hypomagnesemia and a significant increase of arrhythmias and sudden cardiac death, compared to rats with a normal Mg diet [50]. Though intervention studies increasing Mg concentration in drinking water are of interest, given the potential of population-wide intervention, a recent, large population-based cohort study did not demonstrate an association between the incidence of atrial fibrillation and Mg concentration of drinking water [36].

In conclusion, the effects of Mg administration on cardiac function in people with CKD are still undetermined. Yet, based on in vitro data, animal models, and cohort studies, results of Mg intervention studies in relation to cardiac function are logical and could be of value.

### 3.2. Impact of Magnesium Administration on Vascular Disease

Many studies have revealed several mechanisms by which Mg counteracts vascular calcification in vitro. Mg interferes with calcification on a vascular tissue level and influence calcification-promoting and -inhibiting factors [1,19]. In vitro Mg retards calcification of vascular smooth muscle cells (VSMC) and inhibits expression of osteogenic transcription factors [19,60,61,62]. In addition, dietary Mg intervention studies in rodent CKD models confirmed this vascular calcification inhibiting effect of Mg. Several in vivo studies demonstrated reduced development or attenuated progression of arterial calcification when comparing high-Mg diets with low-Mg diets [38,39,40,41,42].

In humans, many studies have observed an inverse association of serum or dietary Mg and the incidence of vascular calcification, including coronary artery calcification (CAC), aortic calcification (AC), and vascular measurements that may partially reflect vascular calcification such as intima-media thickness (IMT) and pulse wave velocity (PWV) [58,63,64,65]. Although Mg intervention studies aiming to improve these outcomes are still relatively limited in number, most are performed within populations that are at high risk of vascular calcification such as people with CKD. A randomized controlled trial (RCT) in people with CKD stage 3–4 investigated the effect of oral Mg oxide supplementation on CAC for two years (starting at 198 mg of elemental Mg per day and titrated to achieve a serum Mg of 1.00–1.23 mmol/L) [25]. This study was prematurely terminated because of a significantly smaller median progression of CAC in the Mg Oxide group (11.3%) compared to the control group (39.5%). In this study, however, no attenuation in changes of the thoracic aorta calcification scores was seen in the Mg oxide group, compared to the placebo (24.2% versus 29.0%, respectively; *p* = 0.89). A pilot study with a sample of seven people on hemodialysis demonstrated no significant change of CAC scores compared to baseline after 18 months of treatment with magnesium/calcium carbonate to achieve a serum phosphate ≤ 1.80 mmol/L (mean dose of 700 mg and 1200 mg elemental Mg and calcium per day, respectively), but did suggest stabilization of the CAC scores [22]. Another pilot study evaluating vascular arterial calcifications measured by X-rays of femur, pelvis, hands, and abdomen in 72 haemodialysis participants compared the effect of Mg carbonate in addition to the phosphate binder calcium acetate versus calcium acetate alone for 12 months [66]. This study observed a difference in progression of arterial calcification (28% vs. 44%, *p* = 0.276) and also a difference in improvement of arterial calcification (16% vs. 0% *p* = 0.04) for the Mg carbonate + calcium acetate group and the calcium acetate group, respectively.

Within hemodialysis populations, three RCTs studied the effect of 2 to 6 months of oral Mg supplementation on IMT [26,29,30]. A recent systematic review and meta-analysis combining these studies demonstrated a significant improvement of IMT for those treated with oral Mg supplements compared to placebo (weighted mean difference (WMD) −0.18; 95% CI −0.34, −0.01) [67]. Of note, no studies addressing IMT as outcome parameter investigated the effect of increasing dialysate Mg concentration for those on dialysis or oral Mg supplementation for people with CKD stage 1 to 4. However, one study investigated the effect of increasing dialysate Mg concentration in relation to PWV in people with end-stage kidney disease [46]. This randomized 4-week crossover study investigated the effect of increasing dialysate Mg from 0.50 mmol/L (regular Mg dialysate concentration) to 0.75 mmol/L on vascular parameters. This study demonstrated a significant decrease (hence, improvement) of PWV of −0.91 m/s (95% CI −1.52, −0.29) for the high-Mg group compared to the regular Mg dialysate concentration [46]. Within obese or hypertensive individuals without CKD, the results of 24 weeks of oral Mg supplementation are conflicting. One RCT in 52 obese individuals demonstrated a significant improvement of PWV by −1.0 m/s (95% CI −0.4, −1.6 m/s) in the Mg citrate compared with the placebo group [68]. However, a very similar RCT comparing different Mg formulas, and another RCT of 24 weeks duration of Mg chelate supplementation in hypertensive women, concluded that there is no significant effect of oral Mg supplementation on PWV [86,87].

A meta-analysis combining nine cohort studies revealed an inverse association between drinking water Mg concentration and mortality due to coronary heart disease (relative risk 0.89; 95% CI 0.79 to 0.99) [37]. To date, however, no study has examined the effect of increasing Mg concentration of drinking water on vascular outcome parameters.

In summary, although limited in number and quite different in design, several studies investigating Mg administration in people with CKD observe small, but likely relevant protective effects in relation to vascular disease, including artery calcification, IMT, and PWV.

### 3.3. Impact of Magnesium Administration on Calcification Propensity

In search for a measure that reflects the inherent calcification propensity of patients’ blood, an in vitro test has been developed recently. The calcification propensity is quantified using a test termed the T50-test. This test challenges patients’ serum with high calcium and phosphate concentrations and follows subsequent transformation of primary calciprotein particles (CPP1) into harmful secondary CPP (CPP2) using light scattering of the sample [88,89]. In clinical settings, observational studies demonstrated that the calcification propensity is an independent predictor of all-cause mortality and CVD in all stages of CKD [88,90,91,92,93]. In vitro experiments demonstrated that Mg dose-dependently delays the transformation of the amorphous CPP1 into the more crystalline CPP2 and thus inhibits phosphate-induced VSMC calcification [62,94]. With each increment of 0.2 mmol/L Mg in serum samples of healthy individuals and participants with CKD, the T50 substantially increased (improved) with 51 ± 15 min and 44 ± 13 min (*p* < 0.05), respectively [62]. These results suggest that inhibition of CPP2 transformation at least partially explains the Mg-mediated protection against vascular calcification. Therefore, several clinical intervention studies investigated the effect of Mg administration on the calcification propensity in different populations [32,44]. After 8 weeks of oral supplementation with twice daily 360 mg Mg hydroxide (30 mmol elemental Mg per day) in people with CKD stage 3–4, the calcification propensity improved significantly compared to the placebo group with 40 min (95% CI 11–70) [32]. In this study, the calcification propensity improvement was already visible at 4 weeks, demonstrating a 30 min longer T50 for those treated with 360 mg Mg hydroxide compared to placebo (95% CI 9–53). A lower dose of Mg supplementation with 360 mg Mg hydroxide once a day (15 mmol elemental Mg per day) did not lead to a significant change of the calcification propensity at 4 and 8 weeks. The same authors examined the effect of increasing dialysate Mg on T50 in people undergoing hemodialysis [44]. After increasing dialysate Mg concentration from 0.5 to 1.0 mmol/L for one month, the T50 increased with 73 min (95% CI 30–116) compared to those with maintained dialysate Mg of 0.5 mmol/L [44].

### 3.4. Impact of Magnesium Administration on Markers of CKD-MBD

Mg is an important factor in CKD mineral bone disorders (CKD-MBD) as it can suppress parathyroid hormone (PTH) secretion, activate the calcium-sensing receptor, promote osteoblast activity, and reduce intestinal phosphate absorption [40,70,71]. Although low Mg levels stimulate PTH secretion, Mg is also required for the production and release of PTH and therefore severe hypomagnesemia can lead to clinical hypocalcemia via hypoparathyroidism [72,95]. For these reasons, administration of Mg is of interest for its potential to modify CKD-MBD. Several intervention studies investigated the effect of Mg administration on these parameters. Of note, in the setting of CKD-MBD, oral Mg has been used as either an intestinal phosphate binder or as dietary supplement with the goal to be absorbed and have systemic effects. Both aspects are shortly discussed here.

Three RCTs compared Mg-containing phosphate binders to other binders in people treated by HD. These studies demonstrated that solely Mg carbonate or a combination tablet of Mg/calcium carbonate/acetate is equally effective in controlling phosphate levels compared to a calcium-only-containing phosphate binder and sevelamer [23,24,69]. However, a recent systematic review and meta-analysis of oral Mg intervention studies in people treated with hemodialysis did not reveal a significant phosphate-lowering effect of these compounds (WMD −0.36 [95% CI −0.89,0.16]) [67]. Most likely, the small phosphate-lowering effect through intestinal chelation of dietary phosphate by oral Mg supplementation is negligible compared to the phosphate clearance during hemodialysis. Additionally, the phosphate-lowering effect in the above-mentioned studies is possibly explained by the phosphate-lowering capacity of the calcium compound in the Mg/calcium combination tablets [24,69]. This would imply that the potential benefit of oral Mg in this setting is not mediated by its small phosphate-binding effect.

The effect of Mg administration on serum calcium and PTH levels have been debated due to conflicting results [24,26,30,44,96,97]. A recent meta-analysis with subgroup analysis for the effect of oral Mg administration demonstrated a significant reduction of both serum calcium levels (WMD −0.50 [95% CI −0.77, −0.23]) and serum PTH levels (WMD −237 [95% CI −350, −123]) [67]. Subgroup analyses for the effect of increased dialysate suggested a comparable yet non-significant effect on serum calcium levels (WMD −2.40 (95% CI −6.91, 2.11)) with no data on PTH levels. Nevertheless, another study on the effect of increasing dialysate Mg observed a significant 21% reduction in PTH levels at the end of intervention compared to baseline for those treated with high (1.00 mmol/L), though compared to the regular Mg dialysate group (0.50 mmol/L) there was no significant difference at the end of the intervention [44].

To summarise, the effects of Mg administration on markers of CKD-MBD are diverse. Mg administration significantly improves serum calcium and PTH, with most evidence for oral Mg interventions. Current studies indicate no clinically relevant phosphate binding-effect of oral Mg supplements.

### 3.5. Magnesium Administration and Other Clinically Relevant Outcomes

Though cardiovascular outcomes and markers of CKD-MBD have been the main focus of Mg intervention studies in CKD populations, other clinically relevant outcomes have also been addressed. These outcomes include inflammation, hemodynamic parameters, endothelial function, metabolic profile, hemodynamic parameters, endothelial function, lipid profile, and muscle cramps.

Both Mg intake as well as serum Mg are inversely associated with the risk of type 2 diabetes, glucose levels, and HbA1c within the general population [45,73,74,75,76]. RCTs investigating the effect of Mg administration in relation to parameters of glucose metabolism demonstrate conflicting results. Two randomized placebo-controlled trials with oral Mg chloride administration during 3 to 4 months in subjects with prediabetes or type 2 diabetes, with decreased serum Mg and normal kidney function, both demonstrated an improvement in insulin sensitivity and fasting glucose [77,78]. This was also confirmed in diabetic haemodialysis patients with improved HbA1c and serum insulin levels after 24 weeks of oral Mg oxide supplementation compared to the placebo group [29]. This same effect on insulin levels and insulin resistance was observed in pre-diabetic people with mild to moderate CKD and hypomagnesemia after 3 months of Mg oxide supplementation [27]. In contrast, another RCT with Mg oxide supplementation for 12 weeks in participants with early-stage diabetic nephropathy (albuminuria without decline in eGFR) and low serum Mg levels did observe a negative effect on insulin resistance in those randomized to the Mg intervention compared to placebo [28].

A recent systematic review and meta-analysis of the effects of Mg administration on lipid profile among people with type 2 diabetes demonstrated a significant reduction of low-density lipoprotein (LDL) [79]. This meta-analysis included two RCTs that were performed in a CKD population with contrasting results [28,29]. An RCT of diabetic hemodialysis patients demonstrated a significant reduction of total cholesterol (−0.30 mmol/L [95% CI −0.56, −0.04]) and LDL (−0.29 mmol/L [95% CI −0.52, −0.05]) for those treated with 250 mg Mg oxide per day for 24 weeks, compared to the placebo group [29]. However, in the RCT with 12 weeks of 250 mg Mg oxide supplementation in participants with early-stage diabetic nephropathy and low serum Mg levels, no significant changes were observed in the lipid profile for the primary analyses [28].

Several other Mg intervention studies focused on the effect of Mg administration on low-grade inflammation parameters such as high-sensitive C-reactive protein (hs-CRP). A meta-analysis on the effect of Mg supplements on CRP in non-CKD populations demonstrated a robust and significant reduction in CRP levels [81]. A cross-sectional, multiple adjusted analysis also confirmed an inverse association between serum Mg levels and CRP in people with CKD [80]. However, only one RCT in people with diabetes treated with hemodialysis observed an effect of Mg administration on hs-CRP (−1.57 mg/L [95% CI −2.06, −1.08]) [29], whereas two other Mg intervention studies in a CKD population did not observe any effect on hs-CRP [26,28]. A post-hoc analysis in people treated with hemodialysis and a focus on different inflammatory markers observed significant reductions in both interleukin 6 (IL-6) and tumor necrosis factor α (TNF-α) in those treated with high dialysate Mg (1.00 mmol/L) compare to regular dialysate Mg (0.50 mmol/L) after 28 days [98].

In addition to the effect of high Mg dialysate concentration on arterial stiffness, the cross-over RCT by Del Giorno et al. [46], including 39 participants with ESRD also focused on the effect of increasing Mg dialysate on the hemodynamic profile and endothelial function. After 6 months of an increased Mg dialysate concentration of 0.75 mmol/L, a statistically significant and likely clinically relevant reduction in systolic blood pressure was observed (−12.96 mmHg [95% CI −24.71, −1.22]), without significant change in diastolic blood pressure [46]. The sparse studies investigating the effects of increasing Mg dialysate concentration on intradialytic hemodynamics were small, short-term (2 to 4 weeks), and had conflicting results [82,99]. Neither of the two RCTs investigating endothelial function observed an effect of increasing dialysate Mg concentration or oral supplementation [26,46].

The inverse association between serum Mg and all-cause mortality has been well established in CKD populations [83]. Yet, inherent to the relatively short-term design of intervention studies and the selective inclusion of participants, the effect of Mg administration on mortality is not yet known.

Although muscle cramps are a well-known symptom of hypomagnesemia, a recent Cochrane review concluded there is no evidence that Mg administration provides clinically meaningful muscle cramp prophylaxis in the general population or in people with CKD [100]. However, in this review, data concerning participants on hemodialysis are limited and no distinction was made between people with hypomagnesemia or normal Mg status.

## 4. Risks of Magnesium Administration

Besides the described potential benefits of Mg administration, some disadvantages may exist as well. Some experimental studies concluded possible impaired bone mineralization and growth restriction due to Mg administration [22,38,101]. The effect on growth restriction, however, was found in studies where the 5/6 nephrectomised rats and Klotho knockout mice were exposed to high dietary Mg from birth onwards, with a growing and maturating skeleton [38,40]. This does not reflect the potential clinical position of Mg supplements in people with CKD, as CKD mostly develops during adult life when the skeleton has matured, and such extreme dietary Mg increases are not feasible to the dose limiting laxatives effects oral Mg administration. Some observational studies that investigated the association between serum Mg and mortality in people with CKD demonstrated a U-shaped curve that suggest that a Mg concentration of approximately 1.1 or 1.2 mmol/L is not favourable [102,103]. Although this U-shaped curve is not confirmed by different observational studies that analysed Mg categories substantially above the reference range [83], we would discourage Mg administration if serum Mg concentrations are exceeding 1.2 mmol/L.

Besides these potential disadvantages, the majority of studies demonstrate the opposite, with normal to high Mg levels being associated with a lower risk of hip fracture and normal or improved bone homeostasis [39,40,96,104]. A recent Japanese, two-year observational study in hemodialysis patients demonstrated an inverse association for Mg concentration and the risk of hip fractures, with an OR 0.96 [0.94–0.99] per 0.1 mmol/L increase of plasma Mg [104]. Although, theoretically, an extreme increase of dialysate Mg is imaginable, only cautious dialysate increases are investigated due to the risk of fast intra- and extracellular electrolyte shifts during as well as between dialysis sessions [44,46,99,105]. Symptomatic hypermagnesemia, generally only seen in people with serum Mg concentrations of 1.6 to 2.0 mmol/L or above, is very rare, usually iatrogenic, and can follow decreased renal excretion, intra- to extracellular compartment shifts, and most frequently from excessive intake [106]. Such moderate to severe hypermagnesemia may induce headache, fatigue, nausea, vomiting, dizziness, cutaneous flushing, drowsiness, confusion, muscle paresis, hypotension, hyporeflexia, and bradycardia [1,106]. As previous stated, in people with declining kidney function, renal Mg excretion diminishes, and therefore hypermagnesemia is most often seen in the elderly or people with CKD in combination with Mg-containing laxatives or antacids [107,108,109,110,111].

Despite the well-known laxative effect of oral Mg supplements, most RCTs conclude that the Mg supplements are well-tolerated. Several RTCs report gastro-intestinal side effects or a slightly higher dropout rate due to diarrhea in the Mg intervention group compared to placebo [23,24,25,26,30]. The laxative effect of Mg supplements is dose-dependent. However, the dosage at which people experience hindering laxative effects highly differs between individuals, and this gastro-intestinal side effect generally reduces after the first weeks of supplementation. To reduce the risk of diarrhea, it is recommended to distribute the total daily dose of Mg with intake twice or three times a day.

## 5. Future Perspectives

Currently, multiple RCTs with a Mg intervention are ongoing in people with CKD stage 3 to 4 and people treated with hemodialysis. Several placebo-controlled RCTs investigate the effect of oral Mg supplementation on markers of vascular calcification in people with CKD stage 3 to 4 [31,112]. The MAGiCAL-CKD study investigates the effect of twice daily 364 mg slow-release Mg (30 mmol/day) hydroxide per day for 12 months, with the primary endpoint of change in CAC score and secondary endpoints including change in PWV, calcification propensity, and other markers of vascular calcification and MBD [112]. The ROADMAP-study investigates the placebo-controlled effect of Mg citrate trice daily (350 mg elemental Mg per day) for 24 weeks, with or without the addition of the open-label phosphate binder sucroferric oxyhydroxide (Velphoro^®^) (1000 mg/day), on the primary endpoint PWV. Secondary endpoints include calcification propensity and several other markers of calcification, MBD, and inflammation. In addition, explorative ^18^F-NaF and ^18^F-FDG Positron Emission Tomography (PET) scans will be performed in a subgroup to evaluate the effects of the intervention on vascular calcification and inflammation, respectively [31]. Two other RCTs investigating the effect of oral Mg citrate supplementation on surrogate markers of kidney and cardiovascular health, calcification propensity, and CPP in people with stage 3b to 4 CKD or treated with hemodialysis are planning to start (ClinicalTrials identifiers: NCT05033054, NCT03565913). Additionally, results of RCTs investigating the effect of increasing dialysate Mg concentration are expected. The MAGIC-HD study, is a randomized, placebo-controlled feasibility study in a hemodialysis population, with a stepwise increase of dialysate Mg from the regular 0.50 mmol/L up to 1.00 mmol/L and explorative endpoints including PWV, 48 h cardiac rhythm registrations, and several markers of vascular calcification and MBD [105]. Another large, cluster randomized clinical trial will investigate cardiovascular-related hospitalization, mortality, and patient-reported muscle cramps comparing higher (0.75 mmol/L) versus lower (0.50 mmol/L) hemodialysate Mg concentration (ClinicalTrials identifier: NCT04079582).

## 6. Conclusions

In people with CKD, Mg administration can significantly decelerate vascular disease, with most evidence for favourable effects on arterial calcification, IMT, and calcification propensity. Mg administration also impacts mineral bone metabolism through improvement of serum calcium as well as PTH levels. However, the effects on glucose metabolism, lipid profile, and inflammation parameters remain inconclusive despite several studies within the CKD population. Overall, these results are hampered by the heterogeneity in Mg interventions, including different routes of administration, different formulations, varying dosage and frequency, the heterogeneity within the CKD population ranging from mild nephropathy to people with dialysis dependent end stage kidney disease, and the heterogeneity in intermediate outcome parameters. The majority of Mg intervention studies focus on oral administration and increasing dialysate Mg, with remarkable absence of studies investigating the effect of increasing Mg through dietary intake or drinking water, though understandably challenging. Mg intervention studies on long-term and cardiovascular disease-specific endpoints are still lacking, and (subgroup) analyses for participants with low magnesium levels would be a valuable addition to present knowledge. Current clinical research demonstrates that Mg administration in people with CKD 1 to 5 is safe, without concerns for severe hypermagnesemia or negative interference with MBD. All results combined indicate a potential for Mg administration in clinical practice as a relatively easy and cheap instrument in the prevention of CVD and metabolic complications of CKD. However, before potential, targeted implementation in clinical nephrology practice, clinical trials with consistent results and focus on long-term clinical relevance are necessary. Upcoming results will substantially contribute to current evidence on the potential of Mg administration in nephrology practice.

## Figures and Tables

**Figure 1 nutrients-15-00547-f001:**
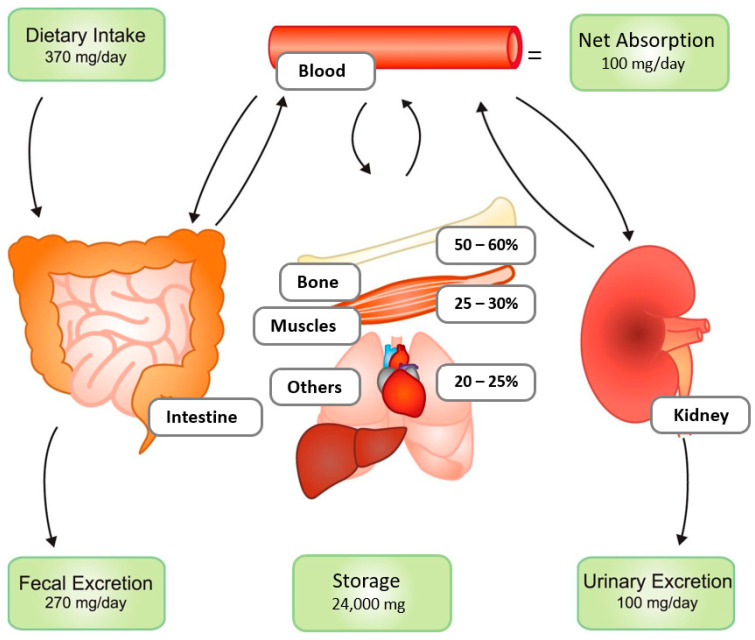
Magnesium (Mg) homeostasis. Panels represent the daily amount of Mg intake and excretion. Daily, the intestines absorb ~120 mg and secrete 20 mg of Mg, resulting in a net absorption of 100 mg. In the kidney daily, ~2400 mg Mg is filtered by the glomerulus, of which 2300 mg is reabsorbed along the kidney tubule. This results in a net excretion of 100 mg, which matches the intestinal absorption. Bone and muscle provide the most important Mg stores. Reproduced with permission [2]. Copyright 2015, the American Physiological Society.

**Table 1 nutrients-15-00547-t001:** Commonly used oral magnesium supplements in intervention studies in people with CKD.

Formulation	Form	Elemental Mg, milligram/day	Elemental Mg, millimol/day	Studies
Mg carbonate	Tablets	700 mg ^a^	29 mmol	[22,23,24]
Mg oxide	Capsules	71 mg ^b^–365 mg	3–15 mmol	[25,26,27,28,29]
Mg citrate	Capsules, pastilles	50 mg ^c^–350 mg	3–15 mmol	[30,31]
Mg hydroxide	Slow release tablets	360–720 ^d^ mg	15–30 mmol	[32]

^a^ Spiegel et al. [22] started at 86 mg elemental Mg per day and up-titrated Mg dosage based on phosphor levels. After up-titration, mean elemental Mg intake was 700 mg per day. ^b^ Mortazavi et al. investigated 250 mg elemental Mg per 3 days. This equals approximately 71 mg per day. Sakaguchi et al. [25] initially administered 198 mg elemental Mg (8.3 mmol) per day and the doses were adjusted every 1 to 3 months to achieve serum Mg levels of 1.03–1.23 mmol/L (2.5–3.0 mg/dL). ^c^ Turgut et al. [30] investigated 98.6 mg elemental Mg every other day. This equals approximately 49.3 mg elemental Mg per day. ^d^ Bressendorff et al. [32] investigated two different Mg intervention regimes, 360 mg elemental Mg once daily (15 mmol Mg) and 360 mg twice daily (equals 720 mg/30 mmol Mg per day).

**Table 2 nutrients-15-00547-t002:** Summary of discussed results that support a rationale for Mg administration in CKD.

	In Vitro	Animal Exp.	In Human Observational (Authors, Ref, Type of Study, Population, Outcome)	In Human Intervention Studies (Authors, Ref, Type of Study, Population, Intervention, Outcome)
Cardiac health				
- Arrhythmia’s	n.a.	✓ [50]	✓ Misialek et al. [51], observational, GP, ↓ serum Mg associated with ↑ AF risk ✓ Khan et al. [52], observational, GP, ↓ serum Mg associated with ↑ AF risk ✓ Tsuji et al. [53], observational, GP ↓ serum Mg associated with ↑ ventricular arrhythmias	✓ Rob et al. [54], RCT, hemodialysis patients, Mg dialysate of 0.25 vs. 0.50 mmol/L, fewer arrhythmias at high Mg dialysate
- Heart failure	n.a.		✓ Reffelmann et al. [55], observational, GP, ↓ serum Mg associated with ↑ LVM ✓ Wannamethee et al. [56], observational, older men, ↓ serum Mg associated with ↑ incident heart failure ✓ Lutsey et al. [57], observational, GP, ↓ serum Mg associated with ↑ incident heart failure	
- Sudden cardiac death	n.a.		✓ Kieboom et al. [58], observational, GP, ↓ serum Mg associated with ↑ increased risk of coronary heart disease mortality and sudden cardiac death ✓ De Roij van Zuijdewijn et al. [59], observational, hemodialysis patients, ↓ serum Mg associated with ↑ cardiovascular mortality and sudden death	
Calcification				
- Vascular calcification	✓ [60,61,62]	✓ [38,39,40,41,42]	✓ Hruby et al. [63], observational, GP, ↓ Mg intake associated with ↑ CAC and AAC ✓ Molnar et al. [64], observational, peritoneal dialysis patients, ↓ serum Mg associated with ↑ AAC ✓ Ito et al. [65], observational, people with CKD, ↑ serum Mg associated with ↓ AAC	✓ Sakaguchi et al. [25], RCT, people with CKD, Mg Oxide vs. placebo, Mg administration reduces CAC progression ✓ Tzanakis et al. [66], RCT, hemodialysis patients, Mg carbonate vs. calcium acetate, Mg administration retards arterial calcifications
- IMT	n.a.	n.a.		✓ Mortazavi et al. [26], RCT, hemodialysis patients, Mg oxide vs. placebo, Mg administration improves IMT ✓ Talari et al. [29], RCT, diabetic hemodialysis patients, Mg oxide vs. placebo, Mg administration improves IMT ✓ Turgut et al. [30], RCT, hemodialysis patients, Mg citrate vs. calcium acetate, Mg administration improves IMT ✓ Guo et al. [67], systematic review and meta-analysis, hemodialysis patients, Mg administration improves IMT
- PWV	n.a.	n.a.		✓ Joris et al. [68], RCT, obese adults, Mg citrate vs. placebo, Mg administration improves PWV ✓ Del Giorno et al. [46], RCT, hemodialysis patients, Mgdialysate of 0.50 vs. 0.75 mmol/L, high Mg dialysate improves PWV
- T50	✓ [62,69]			✓ Bressendorff et al. [32], RCT, people with CKD, Mg hydroxide vs. placebo, Mg administration improves T50 ✓ Bressendorff et al. [44], RCT, hemodialysis patients, Mgdialysate of 0.50 vs. 1.00 mmol/L, high Mg dialysate improves T50
Markers of CKD-MBD				
- Phosphate		✓ [40]		✓ Tzanakis et al. [23], RCT, hemodialysis patients, Mg carbonate vs. calcium carbonate in combination with a dialysate Mg of 0.30 mmol/L, comparable phosphate, phosphate x calcium product and PTH and better calcium levels ✓ Spiegel et al. [24], RCT, hemodialysis patients, Mg carbonate vs. calcium acetate, equal control of serum phosphorus, lower calcium ingestion ✓ Bressendorff et al. [44], RCT, hemodialysis patients, Mg ✓ dialysate of 0.50 vs. 1.00 mmol/L, high Mg dialysate decreases serum phosphate ✓ De Francisco et al. [69] RCT, dialyses patients, calcium acetate/Mg carbonate vs. sevelamer, non-inferiority of calcium acetate/Mg carbonate for phosphate levels
- PTH	✓ [70]	✓ [40]	✓ Navarro et al. [71], observational, hemodialysis patients, ↑ serum Mg associated with ↓ PTH	✓ Rude et al. [72], RCT, hyperparathyroid patients, and GP, Mg administration, Mg administration results in PTH increase in hypomagnesemic patients and PTH decreases in hyperparathyroid patients ✓ Bressendorff et al. [44], RCT, hemodialysis patients, Mgdialysate of 0.50 vs. 1.00 mmol/L, high Mg dialysate decreases PTH ✓ Guo et al. [67], systematic review and meta-analysis, Mg administration, Mg administration reduces PTH
Other clinically relevant outcomes				
- Glucose metabolism	n.a.		✓ Dong et al. [73], meta-analysis, GP, ↑ dietary Mg intake associated with ↓ risk of T2DM in a dose-response manner ✓ Ma et al. [74], observational, GP, ↑ dietary Mg intake is associated with ↓ fasting serum insulin ✓ Hruby et al. [75], observational, GP, ↑ dietary Mg intake is associated with ↓ incident T2DM ✓ Kieboom et al. [76], observational, GP ↓ serum Mg levels are associated with ↑ risk of prediabetes T2DM	✓ Rodriguez-Moran et al., [77], RCT, hypomagesemic T2DM and Mg chloride, Mg administration improves insulin sensitivity and metabolic control ✓ Guerrero-Romero et al. [78], RCT, pre-diabetic and hypomagnesemic patients, Mg chloride, Mg administration improves glucose levels and glycemic status ✓ Toprak et al. [27], RCT, pre-diabetic, obese people with CKD, Mg oxide, Mg administration improves insulin sensitivity and metabolic control ✓ Talari et al. [29], RCT, diabetic hemodialysis patients, Mg oxide, Mg administration improves insulin metabolism and HbA1c
- Lipid metabolism	n.a.			✓ Asbaghi et al. [79], systematic review and meta-analysis, T2DM, Mg administration, Mg administration improves LDL ✓ Talari et al. [29], RCT, diabetic hemodialysis patients, Mg oxide, Mg administration improves LDL and total cholesterol
- Low grade inflammation			✓ Liu et al. [80], observational, hemodialysis patients, ↓ serum Mg is associated with ↑ CRP levels	✓ Mazidi et al. [81], systematic review and meta-analysis, diverse populations, Mg administration, Mg administration reduces CRP levels ✓ Bressendorff et al. [45], RCT, Mg dialysate of 0.50 vs. 1.00 mmol/L, high Mg dialysate reduces systemic inflammation ✓ Talari et al. [29], RCT, diabetic hemodialysis patients, Mg oxide, Mg administration improves high-sensitive CRP levels
- Blood pressure	n.a.	n.a.		✓ Kyriazis et al. [82], RCT, hemodialysis patients, 0.75 dMg, 1.75 dCalcium (group I); 0.25 dMg, 1.75 dCalcium (group II); 0.75 dMg, 1.25 dCalcium (group III); 0.25 dMg, 1.25 dCalcium (group IV), ↑ dMg decreases the incidence of intradialytic hypotension ✓ Del Giorno et al. [46], RCT, hemodialysis patients, Mg dialysate of 0.50 vs. 0.75 mmol/L, high Mg dialysate improves systolic blood pressure
Mortality				
Cardiovascular Mortality	n.a.		✓ Leenders et al. [83], systematic review and meta-analysis, people with CKD, ↑ serum Mg associated with ↓ incidence of cardiovascular mortality and cardiovascular events ✓ Kieboom et al. [58], observational, GP, ↓ serum Mg associated with ↑ risk of coronary heart disease mortality and sudden cardiac death ✓ Jiang et al. [37], meta-analysis, GP, ↑ drinking water Mg associated with ↓ coronary heart disease mortality	
All-cause mortality	n.a.		✓ Leenders et al. [83], systematic review and meta-analysis, people with CKD, ↑ serum Mg associated with ↓ all-cause mortality	

Abbreviations: n.a., not applicable; AAC, abdominal aortic calcification; AF, atrial fibrillation; CAC, coronary artery calcification; CKD, chronic kidney disease; CRP, c-reactive protein; d, dialysate; exp, experiments; GP, general population; IMT, intima-media-thickness; LDL, low density lipoprotein; LVM, left ventricular mass; MBD, mineral bone disease; Mg, magnesium; PTH, parathyroid hormone; PWV, pulse wave velocity; T2DM, type 2 diabetes mellitus; T50, calcification propensity blood test; vs., versus; ↑, high, higher or increased; ↓, low, lower or decreased.

## Data Availability

Not applicable.

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
