# Peer review of "Magnesium Administration in Chronic Kidney Disease"

_nutrients, 2023, doi:10.3390/nu15030547_

Round 1
Reviewer 1 Report
Vermeulen and Vervloet have submitted a very well-written review on Mg supplementation in CKD.
The topic is timely and of significance as the potential benefits of Mg supplementation are currently being actively investigated.
The authors cover the pertinent aspects very well.
The current status of the field is excellently clarified and areas for further investigation are also covered well.
This reviewer has only a few minor suggestions
1. Are there data about what the "usual" Mg concentration is at all stages of CKD? Is it any different than individuals who do not have CKD?
2. A table describing the potential benefits of Mg supplementation or increasing Mg levels on heart, vasculature, etc coupled with an indication of whether the data are primarily in vitro, in vivo, or both, and what these data show would be great for the reader to provide an overview of what we know now.
3. The authors have very little description about the mechanisms at a more molecular level. Recognizing that the purpose of this review was not to discuss exhaustively the effects of Mg at a molecular level, it is still reasonable at least to touch on these mechanisms.
4. The authors discuss potential risks of Mg supplementation. Is there a safe "high" level of Mg to target for non-CKD and CKD patients?
5. Few typos noted that would not be picked up by spell-check
page 3, line 71 -- form should be from
page 4, line 156 -- proof should be prove
page 6, line 253 -- an should be and
page 7, line 279 -- censing should be sensing
page 10, line 437 -- save should be safe
Author Response
Thank you fore your time and suggestions. Please see the attachement for our point by point reply.
Kind regards,
Emma Vermeulen

Reviewer 2 Report
Vermeulen and Vervloet presented a narrative review to discuss the possible health benefits of magnesium interventions in patients with chronic kidney disease. Despite the fragmentary data available, it emerges that magnesium can have health benefits, in the face of limited, mostly reversible or modulable side effects such as diarrhoea.
The topic is fascinating, especially the introduction which shows a complete and exhaustive background about the role of magnesium and the possible modulation mechanisms in the body compartments. The collection of intervention data is more fragmented, perhaps due to limited sources. Still, the use of in vitro and preclinical data, as well as data on individuals without chronic kidney disease, emerges as offtopic on the role of magnesium in patients with CKD.
In particular:
- If the topic is focused on the effect of magnesium on patients with CKD, I advise the authors to review only intervention studies consistent with this topic (e.g. pages 5-8). No preclinical or healthy subjects
- The term "supplementation" could be misleading. If authors are not discussing exclusively oral magnesium, it would be better to talk about administration, delivery or similar, changing the title. I do not think that increasing magnesium concentrations in the dialysate can be considered supplementation.
- In lines 47-49, the relationship between magnesium and vitamin D should be better clarified
- It would be better to remove lines 92-94. Even if the manuscript is not focused on circulating Mg concentrations, it is still an aspect that cannot be ignored and the authors suggest how is important an assessment in intervention studies
- It could be useful to investigate the rationale of all the magnesium formulations. The authors explain Mg carbonate and Mg citrate but do not dwell on the other two
- CAC is discussed on page 5. These parts could be moved to the next paragraph, for greater consistency
- Line 208 refers to a figure not present in the manuscript
- The conclusions should be more concise, for example, the discussion of ongoing trials could be moved to a dedicated paragraph before the conclusions
Minor aspects:
- At line 8 the concept of reverse association is not clear if it is not precisely established what the authors refer to (concerning adverse outcomes?)
- At line 17 does “save” mean “safe”?
- Some spacing would be needed between line 115 and line 116
- On lines 134 and 358 there are font homogeneity problems
- At 255 a space is missing before references 74-75.
- A comma is missing at line 320
- Move the bibliographic reference immediately after the author from line 366 to line 361
- If there are no supplementary materials, lines 471-472 should be removed
Author Response

(The authors gave the same response as above.)

Round 2
Reviewer 2 Report
I suggest implementing table 2 (there is a typo about numbering) including the main characteristics (type of study, host, intervention, etc. ) and results of the studies.
Author Response
We thank the reviewer for the opportunity to extend on the information provided in Table 2. “Summary of discussed results that support a rationale for Mg administration in CKD”. We added the following information for each reference in the column of ‘in human studies’: authors, reference, type of study, population, (intervention) and outcome. Since it this is a review on magnesium administration in human, we did only elaborate on the additional information of in human studies and did not further extend the table with information on non-clinical evidence.
With these additional specifications of each in human study, we trust that we provided the reader with a sufficient overview of the potential benefits of magnesium administration on all relevant outcomes that we discuss in paragraph 3.1-3.6.
We apologize for not finding a typo about numbering of this table. The below reference numbering does now match the reference numbering of the manuscript if that was the issue the reviewer was referring to.
